# Modulatory Action of Insulin-like Growth Factor I (IGF-I) on Cortical Activity: Entrainment of Metabolic and Brain Functions

**DOI:** 10.3390/cells14171325

**Published:** 2025-08-27

**Authors:** Nuria García-Magro, Alberto Mesa-Lombardo, Ángel Nuñez

**Affiliations:** 1Department of Anatomy, Faculty of Health Science, Universidad Francisco de Vitoria, Pozuelo de Alarcón, 28223 Madrid, Spain; nuria.garcia@ufv.es; 2Department of Anatomy, Histology and Neurosciences, Universidad Autónoma de Madrid, 28029 Madrid, Spain; alberto.mesa@estudiante.uam.es

**Keywords:** IGF-I, growth hormone, cholinergic neurons, orexinergic neurons, synaptic plasticity, electrocorticogram, circadian rhythms

## Abstract

Insulin-like growth factor I (IGF-I) is a neurotrophic factor that regulates neurogenesis, synaptogenesis, and neuronal survival. It also enhances neuronal activity and facilitates synaptic plasticity. Additionally, IGF-I plays a critical role in the regulation of metabolism in mammals. Emerging evidence indicates that IGF-I modulates sleep architecture. The circadian integration of metabolic and neuronal systems serves to optimize energy utilization across the light/dark cycle. Current data suggest that IGF-I may be a key mediator of this integration, promoting brain activity during wakefulness, a state that coincides with increased metabolic demand. In this review, we summarize recent findings on the interplay between metabolism, IGF-I, and brain activity.

## 1. Introduction

Brain activity is in continuous variation, influencing not only neural responses to sensory inputs but also the ability to process information and make decisions. Numerous investigations have provided electrophysiological, pharmacological, and anatomical evidence demonstrating that cholinergic inputs from the basal forebrain, noradrenergic inputs from the locus coeruleus, and serotonergic inputs from the raphe nuclei to the thalamus and cortex, independently or collectively, produce a neuronal activation pattern during the waking state. This activation pattern is reflected in the appearance of fast oscillatory activity in the electrocorticogram (ECoG) [1]. In addition, neural networks exhibit a high degree of interconnectivity during wakefulness, facilitating information processing in the central nervous system (CNS). Other neuromodulators of brain activity are also engaged during this state, contributing to elevated levels of neuronal activity. One such neuromodulator is insulin-like growth factor I (IGF-I).

IGF-I is a neurotrophic factor that participates in multiple cellular processes within the CNS. It is a 70-amino acid polypeptide arranged in a single chain, and its production is mainly controlled by growth hormone (GH) [2]. Most circulating IGF-I is produced by the liver, with smaller amounts originating from adipose tissue and muscle [3], and it reaches the brain through the blood–brain barrier or via the choroid plexus [4,5].

One of the most important functions of IGF-I is the modulation of brain activity. IGF-I plays a crucial role in the growth and development of the CNS by mediating the effects of GH. Although most IGF-I enters the brain via the blood–brain barrier or the choroid plexus, it is also produced by all major CNS cell types, particularly in the neocortex, hippocampus, cerebellum, and hypothalamus [6]. In the adult brain, IGF-I participates in brain development by regulating cell proliferation, differentiation, survival, neuronal plasticity, and response to injury [7,8,9]. In the adult brain, IGF-I also plays an essential role in modulating synaptic activity and is a key factor in the control of cognitive functions [10].

## 2. IGF-I Signal Pathways

Since 1986, when Ullrich et al. first published the primary structure of IGF-I receptor (IGF-IR) [11], the presence of IGF-IR has been extensively demonstrated in different cell types and tissues [7,12,13,14]. IGF-IR has a high affinity for IGF-I (IC_50_: 0.2–0.8 nM) and IGF-II (IC_50_: 0.5–4.4 nM), but it can also bind insulin, albeit with a 50- to 100-fold lower affinity (IC_50_: >30 nM). IGF-IR is a disulfide bond-linked tetramer composed of two extracellular α subunits, which bind specific ligands, and two transmembrane β subunits, which possess tyrosine kinase activity [15]. Ligand binding to the α-subunits induces a conformational change in the β-subunits, resulting in activation of the receptor’s tyrosine kinase domain and autophosphorylation of specific tyrosine residues, thereby initiating downstream signaling processes. IGF-I and insulin receptors share 48% amino acid sequence homology and can form functional hybrid receptors capable of recognizing both ligands [16].

The canonical signaling pathways of IGF-IR are shared with those of the insulin receptor and have been described in most cell types. These include the PI3K/AKT and Ras/MAPK pathways [16,17]. Activated PI3K/AKT pathway recruits AKT to the plasma membrane by interaction with second messengers, where it is phosphorylated and activated. Activated AKT subsequently phosphorylates several downstream targets, such as Bad, Caspase-9, and mTOR, promoting cell survival and growth [18]. The Ras/MAPK pathway is an important signaling pathway that regulates a wide range of cellular processes such as proliferation, differentiation, apoptosis, and stress responses [19]. In addition, other non-canonical IGF-IR signaling pathways have been described [13], but these are beyond the scope of this review (Figure 1).

## 3. IGF-I and Its Role in Metabolism and Mitochondrial Function

IGF-I is a critical anabolic hormone that influences growth, metabolism, and cellular survival. Beyond its well-known role in development, IGF-I also plays a significant role in regulating glucose transport and mitochondrial function, making it essential for maintaining energy homeostasis and cellular health. Although IGF-I is closely related to insulin, the two hormones have distinct metabolic actions. Insulin primarily regulates carbohydrate, fat, and protein metabolism. In muscle tissue, IGF-I stimulates glucose uptake—similarly to insulin—by enhancing the transport of glucose across cell membranes through the glucose transporter type 4 (GLUT4) [20].

Upon binding to the IGF-IR, it stimulates the phosphoinositide 3-kinase (PI3K)/Akt pathway, leading to the translocation of GLUT4 to the cell membrane. This mechanism is particularly important in skeletal muscle and adipose tissue [21]. Moreover, IGF-I enhances insulin sensitivity and supports glucose utilization, thereby contributing to metabolic flexibility. In neurons, IGF-I facilitates glucose transport by upregulating the expression of the glucose transporter type 1 and 3 (GLUT1 and GLUT3), which is critical for brain metabolism and function [22]. In addition, IGF-I is a key stimulator of protein synthesis in muscle via the PI3K-Akt-mTOR pathway and inhibits proteolysis, leading to muscle hypertrophy. In the liver and adipose tissue, IGF-I promotes lipogenesis and glucose storage [7,20].

Differences in receptor distribution across cell types may account for many of the functional distinctions between insulin and IGF-I. Mature hepatocytes and adipocytes express abundant insulin receptors but have minimal IGF-I receptor expression. In contrast, vascular smooth muscle cells have a high density of IGF-I receptors and relatively few insulin receptors. Despite these differences, IGF-I regulates both the growth and development of the entire organism and specific differentiated functions in individual tissues. In general, IGF-I lowers blood glucose levels, stimulates protein synthesis, and reduces proteolysis. The effect of IGF-I on proteolysis is most likely mediated through activation of the insulin receptor, as relatively high doses of IGF-I are required to elicit this response, and insulin itself has been shown to inhibit proteolysis [23].

IGF-I also exerts a significant influence on mitochondrial function. It stimulates mitochondrial biogenesis through peroxisome proliferator-activated receptor gamma coactivator 1-alpha (PGC-1α), Akt and mTOR pathways [24]. As a result, mitochondrial DNA replication is enhanced, followed by the synthesis of mitochondrial proteins, both crucial for sustaining cellular energy production.

In addition to promoting mitochondrial quantity, IGF-I also enhances mitochondrial quality by improving oxidative phosphorylation efficiency and ATP synthesis, particularly in the muscle and brain. Furthermore, it reduces the production of reactive oxygen species (ROS) by increasing antioxidant enzyme activity such as superoxide dismutase and catalase [25]. In neuronal cells, IGF-I helps maintain mitochondrial membrane potential, reduces apoptosis, and supports synaptic energy demands, thereby potentially protecting against neurodegenerative conditions [26]. Together, these actions position IGF-I as a key modulator of both glucose metabolism and mitochondrial integrity, with implications for aging, metabolic disease, and neuroprotection.

## 4. IGF-I as a Modulator of Brain Activity

The insulin superfamily of peptides, an evolutionarily conserved group, has emerged as fundamental to CNS growth and development [7,27]. IGF-I is a potent growth factor in the CNS, exerting pleiotropic effects on all major cellular types. IGF-IRs appear prior to brain formation, initially in the ventral floor plate. In the adult brain, these receptors are widely distributed across various regions, including the neocortex, hippocampus, and spinal cord [28].

Specifically, the action of IGF-I in the brain, as in other tissues, is mainly mediated by PI3K/AKT and Ras/MAPK pathways [13]. Activation of the PI3K-Akt pathway affects the activity of different targets such as rapamycin (mTOR), glycogen synthase kinase 3β (GSK-3β), and β-catenin [29], which are involved in membrane trafficking of neurotransmitter receptors [30]. In the Ras-Raf-MAPK pathway, IGF-I rapidly induces phosphorylation and activation of ERK1/2 and p38 MAPK, which are crucial for cellular maturation and survival [29].

As indicated above, the circulating plasma IGF-I enters the brain via the blood–brain barrier or the choroid plexus, suggesting a tonic transfer of serum IGF-I into the brain [4,22]. Circulating IGF-I has been shown to enter the brain following activation of IGF-IRs and through the membrane multicargo transporter megalin (low-density lipoprotein receptor-related protein 2 (LRP2) [31]. Neurons and glial cells are also capable of synthesizing IGF-I within the CNS [32,33]. While a tonic supply of IGF-I from the serum has been proposed, its transport into the CNS is actively enhanced during periods of elevated neuronal activity or physical exercise. [5,22,34]. Indeed, electrical stimulation of the cerebellar peduncle evokes an increase in cerebellar IGF-I concentration, and whisker stimulation increases IGF-I concentration in the primary somatosensory cortex [22]. Therefore, different brain regions may exhibit different IGF-I levels depending on their neuronal activity. In line with these observations, the increase in IGF-I-immunoreactive neurons in the developing rat visual cortex following experience-dependent neuronal activity (Ciucci et al., 2007) [35], or after environmental enrichment [36], may be explained by the localized transfer of serum IGF-I into the brain. These findings are further supported by evidence that human subjects with higher serum IGF-I levels exhibit increased brain activity [37].

IGF-I plays a variety of actions in the CNS. It enhances neuronal and glial activity, controlling numerous brain processes [7,38,39,40,41,42,43]. Indeed, IGF-I increases action potential firing and the response to synaptic inputs in cortical neurons [5,44,45,46,47,48]. For example, local application of IGF-I to the primary somatosensory cortex enhances tactile responses in mice (Figure 2A), or increases the receptive field in the primary somatosensory cortex of rats [49]. Additionally, IGF-I enhances glutamatergic synaptic transmission in hippocampal slices from juvenile rats through activation of PI3K [50,51], as well as in the neocortex of both young adult and aged rodents [42,44,52,53,54]. Other authors have shown that IGF-I reduces the amplitude of EPSCs in the prefrontal cortex by activating metabotropic receptors and inducing AMPA receptor endocytosis [55]. Therefore, IGF-I can modulate neuronal activity depending on the properties of the neurons within each brain region.

One of the most important functions in the CNS is synaptic plasticity, which enables the modulation of the brain response to the characteristics of the stimulus and the behavioral context. Evidence from human and animal studies indicates that IGF-I is a crucial regulator of synaptic plasticity, which is central to learning and memory processes. Sensory experience or repetitive synaptic inputs can induce long-term potentiation or depression (LTP and LTD) in cortical neurons through changes in intrinsic excitability as well as structural alterations, including the formation, elimination, and morphological remodeling of cortical synapses and dendritic spines [56]. LTP, first discovered in hippocampal synapses [57], is a fundamental mechanism for synapses to store new information. Multiple forms of LTD exist and may play different roles in plasticity by reducing synaptic inputs [58]. Given the important role that synaptic plasticity plays in information processing and various other cognitive functions, the modulation of synaptic plasticity by different neurotransmitters, such as IGF-I, is highly relevant.

The involvement of IGF-I in modulating synaptic plasticity is well established in many systems. LTP has been observed between granule and mitral cells in the olfactory bulb, and this LTP requires activity-dependent IGF-I signaling for odor memory formation [59]. In anesthetized rodents, repetitive stimulation of the whiskers induces LTP in the primary somatosensory cortex of both rats and mice. Figure 2B shows an example where LTP is enhanced by the application of IGF-I on the cortex. This facilitation of LTP is attributed to the activation of NMDA receptors [44,53,60,61]. It has also been shown that high (10 nM) concentrations of IGF-I induce LTP in in vitro recordings of pyramidal cortical neurons, whereas lower (7 nM) concentrations of IGF-I induce LTD [42]. IGF-I is crucial for hippocampal LTP and the acquisition of new spatial memories, as demonstrated in a conditional IGF-I knockout model where brain IGF-I synthesis is severely reduced [62]. These findings suggest that brain levels of IGF-I may be crucial for the modulation of synaptic plasticity. Consistent with this, chronic adult-onset GH/IGF-I hypersecretion, caused by tumors in rats, increases LTP in the dentate gyrus of the hippocampus [63].

Data from human studies indicate that higher circulating IGF-I levels are positively correlated with enhanced cognitive abilities. Higher serum IGF-I concentrations have been associated with improved performance in tasks involving memory, attention, and executive function, suggesting that IGF-I contributes to maintaining cognitive functions across the lifespan) [6,37]. In contrast, cognitive impairments have been observed in patients with GH/IGF-I deficiency [64,65,66]. Likewise, mice with lower serum IGF-I levels show cognitive deficits accompanied by impaired hippocampal LTP [67,68].

## 5. IGF-I and Cortical Activation

The above findings indicate that IGF-I modulates both the intrinsic and synaptic properties of cortical neurons. These effects on neuronal function are reflected at the network level in the population activity recorded through the electrocorticogram (ECoG). Indeed, IGF-I administration enhances fast-frequency oscillations in the ECoG of both mice and non-human primates [69,70]. In contrast, mice lacking functional IGF-IRs exhibit increased slow delta wave activity in the ECoG [47]. In humans, lower serum IGF-I levels in middle-aged individuals are associated with reduced coherence in beta and theta frequency bands of the EEG [71]. Thus, IGF-I enhances cortical activity by promoting fast-frequency oscillations, thereby facilitating information processing. This effect may result not only from the direct action of IGF-I on cortical neurons, but also from its influence on modulatory systems that support wakefulness, such as the basal forebrain cholinergic system and the hypothalamic orexinergic system.

Electrophysiological recordings in the basal forebrain (BF) indicate that cortical activation is mainly dependent on BF projections [72,73,74]. These effects are largely mediated by acetylcholine release in the cortex during both wakefulness and rapid eye movement (REM) sleep [75,76]. Intraperitoneal injections of IGF-I enhance the activity of optogenetically identified cholinergic BF neurons and induce elevated cFos expression in these cells. Furthermore, IGF-I application in the horizontal limb of the diagonal band induced fast (>4 Hz) oscillations in the ECoG [77].

Orexin neurons (also termed hypocretin neurons) are located in the perifornical (PeF) area of the lateral hypothalamus [78,79]. They are crucial in regulating various physiological functions, including the sleep/wake cycle [80] or physical activity [81]. These neurons receive multiple endocrine modulatory inputs, such as insulin and IGF-I [82]. We have shown that optogenetically identified orexin neurons are activated by IGF-I, thereby facilitating wakefulness and ECoG activation [47]. Altogether, these data highlight that IGF-I plays a crucial role in modulating neuronal activity and synaptic plasticity across multiple brain regions.

## 6. The Role of IGF-I in Cognitive Impairment and Neurodegenerative Disorders

While IGF-I generally promotes brain activity, its effects may be disrupted under pathological conditions characterized by chronically reduced circulating IGF-I. Conversely, some studies have found that elevated IGF-I levels can be linked to impaired cognitive function. For this reason, several clinical and epidemiological studies have investigated the involvement of IGF-I in neurodegenerative diseases such as Alzheimer’s disease (AD), Parkinson’s disease (PD), and in healthy aging conditions in which a gradual decline in both circulating and brain IGF-I levels has been reported [10].

Animal models with IGF-I deficiency have exhibited deficits in LTP and spatial learning, reinforcing the essential role of this factor in synaptic plasticity and memory processes [62,68]. As noted above, low serum IGF-I levels in humans have been associated with diminished cognitive performance, including reduced fluid intelligence and memory function, particularly in elderly individuals [37,83]. Indeed, subjects with GH/IGF-I deficiency—such as those with hypopituitarism—frequently present cognitive impairments, reduced mental well-being, and a higher incidence of psychiatric symptoms [65,66]. Below, we present examples of IGF-I involvement in neurodegenerative diseases. We focus on AD and PD because they are the two most common neurodegenerative disorders, whose prevalence is increasing alongside global population aging.

### 6.1. The Role of IGF-I in the Pathophysiology of Alzheimer’s Disease

AD is characterized by memory loss, typically accompanied by a decline in visuospatial and executive functions [84]. The two main neuropathological hallmarks of AD are extracellular plaques formed by the abnormal accumulation of β-amyloid (Aβ), and intracellular fibrillary tangles composed of hyperphosphorylated tau, a microtubule-associated protein [85,86]. The accumulation of these proteins is also associated with microglial activation and a pro-inflammatory environment, ultimately leading to hippocampal and cortical neurodegeneration [87,88,89]. Given IGF-I’s neuroprotective properties, it has been proposed that reduced IGF-I levels and decreased sensitivity of IGF-IR in the brain contribute to increased vulnerability to AD [90,91]. Experimental studies suggest a link between IGF-I dysfunction and the risk and pathology of AD. Synapse loss, for instance, is recognized as a primary structural alteration underlying cognitive decline in AD [92]. IGF-I plays a crucial role in physiological synaptogenesis both during development and in the adult brain [93]. Thus, the reduced levels of IGF-I in AD may contribute to the characteristic memory loss of this disease due to a decrease in synaptic transmission.

Moreover, increased seizure susceptibility observed in AD patients [94,95] has also been reported in mice with serum IGF-I deficiency [48]. In addition, disruptions in the sleep/wake cycle are commonly observed in AD, probably, at least in part, due to a reduction in IGF-I levels since it has strong excitatory effect on orexinergic neurons [47,48].

### 6.2. IGF-I in Parkinson’s Disease

In Parkinson’s disease (PD), the pathological hallmark is the accumulation of insoluble α-synuclein aggregates (Lewy bodies) within the cytoplasm of dopaminergic neurons in the substantia nigra pars compacta, which leads to neuronal degeneration [84]. This degeneration results in the classic motor symptoms of PD, such as bradykinesia, rigidity, and tremor. In humans, IGF-I levels are frequently reduced in individuals with PD, and this reduction has been associated with greater disease severity and motor dysfunction [96,97,98]. Furthermore, cerebrospinal fluid levels of IGF-I may also be altered in PD, although findings are still inconsistent, possibly due to differences in disease stage or treatment effects.

Experimental studies in animal models of PD have indicated that IGF-I can exert multiple neuroprotective effects. IGF-I has been shown to reduce dopaminergic neuron loss, mitigate neuroinflammation, and support mitochondrial integrity [33]. IGF-I also exerts a protective effect against α-synuclein aggregation, helping to maintain the survival of developing neurons and protect mature neurons from excitotoxic damage [99], promote mitochondrial health, and improve mitochondrial biogenesis and function, which are crucial since mitochondrial dysfunction is a key pathogenic mechanism in PD [100]. Although causality is still under investigation, these findings suggest that IGF-I may play a role in modulating the progression of PD.

## 7. IGF-I and the Regulation of the Sleep–Wake Cycle

IGF-I plays a crucial role in various physiological processes, including growth, metabolism, and brain function. Emerging evidence indicates that IGF-I contributes to the regulation of the sleep–wake cycle, primarily through its interactions with various CNS structures, particularly the circadian clock. The production of IGF-I is stimulated by GH, and many of the effects of GH are mediated through IGF-I [101]. Pituitary GH secretion is regulated by two hypothalamic proteins: growth hormone-releasing hormone (GHRH), which stimulates GH release, and somatostatin, which inhibits it. The peak of GH secretion occurs mainly during deep slow-wave sleep, indicating that there is a correlation between GH secretion, and therefore IGF-I levels, and the sleep–wake cycle [102,103].

The circadian clock orchestrates endogenous rhythms in physiological processes and behavior, collectively referred to as circadian rhythms. Through these interactions, IGF-I may influence both the timing and quality of sleep, highlighting its potential role in maintaining proper sleep–wake homeostasis [104,105]. In mammals, the circadian integration of metabolic and neuronal systems optimizes energy utilization throughout the light–dark cycle [106]. Cellular circadian clocks are entrained by the light–dark cycle to synchronize biological processes. These processes underlie various circadian rhythms, including the 24 h sleep–wake and fasting–feeding cycles. Consequently, light–dark information is integrated with energy status to align brain activity with feeding behavior. Circadian rhythms are driven by the hypothalamic suprachiasmatic nucleus (SCN) that is considered the master pacemaker [107]. The SCN regulates endocrine and metabolic rhythms by modulating nocturnal secretions of prolactin and GH, as well as by controlling the rhythmic release of hormones such as melatonin and cortisol. GH secreted by the pituitary gland acts on peripheral organs and stimulates the production of IGF-I [101,103]. Some evidence suggests that IGF-I can interact with core circadian clock genes and influence their expression, further supporting its role in circadian regulation [105].

IGF-I exhibits higher serum levels during the sleep period [108]. In agreement with that, extended sleep duration in rats results in increased IGF-I levels in plasma, muscle, and cortex [109,110]. As previously mentioned, circulating IGF-I enters the brain in response to physical activity [5], or through an activity-dependent mechanism [22]. In fact, systemic administration of IGF-I induces low-voltage fast oscillations in the ECoG of mice and non-human primates [70].

In humans, serum IGF-I concentrations are typically higher in the morning and lower at night [111]. However, adequate sleep in mammals enhances IGF-I circulation, whereas sleep deprivation leads to reduced IGF-I levels in both the brain and muscles [110]. Consequently, sleep deprivation or fragmentation can lower both GH and IGF-I concentrations, potentially impairing tissue repair, metabolism, and cognitive function [103,112]. Although IGF-I levels are lower during wakefulness compared to the sleep period, its uptake by the brain may increase due to heightened physical and neural activity [5,22] and/or feeding schedule [113] (Figure 3).

In addition, IGF-I may synchronize neuronal and metabolic systems through its action on orexin neurons, which are involved in promoting and maintaining arousal [80,114,115] as well as regulate feeding behavior [116]. Selective reduction in IGF-IR signaling in orexin neurons leads to lower hypothalamic orexin levels, elevated slow-wave activity in the ECoG, and reduced sleep-onset latency in mice [47]. Disruption of IGF-IR signaling in orexin neurons also results in altered circadian glucose rhythms and feeding behavior, preceding the emergence of metabolic dysfunction [117]. These findings suggest that IGF-I may contribute to the synchronization of metabolic and circadian processes by modulating orexin neuron activity.

## 8. Conclusions

IGF-I plays essential roles in growth, metabolic processes, and neural function, and participates in the regulation of circadian sleep–wake rhythms. The circadian system, governed by the SCN, coordinates endocrine and metabolic rhythms with the light–dark cycle, including the release of GH and, consequently, IGF-I. Although IGF-I levels are lower during wakefulness, its uptake by the brain may increase due to heightened physical and neuronal activity. Adequate sleep promotes IGF-I circulation, whereas sleep deprivation reduces IGF-I levels in the brain and muscles, potentially impairing tissue repair, metabolism, and cognitive function.

IGF-I may also synchronize neural and metabolic systems by acting on orexinergic neurons in the lateral hypothalamus, which are crucial for maintaining arousal and regulating feeding behavior. Mice with impaired IGF-I receptor signaling in orexinergic neurons show reduced hypothalamic orexin levels, increased slow-wave activity, shorter sleep-onset latency, and disrupted glucose rhythms and feeding patterns. Therefore, these findings suggest that IGF-I contributes to the coordination of metabolic and circadian processes through the modulation of orexin neurons.

## Figures and Tables

**Figure 1 cells-14-01325-f001:**
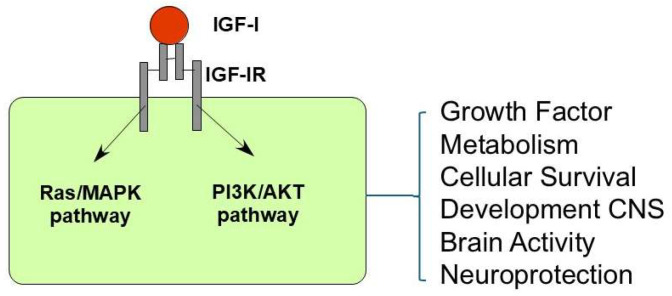
Intracellular signaling mechanisms of IGF-I. IGF-I elicits cellular effects by two principal intracellular signaling pathways: the Ras/mitogen-activated protein kinase (MAPK) pathway and the phosphoinositide 3-kinase/protein kinase B (PI3K/AKT) pathway. Ligand binding to IGF-IR leads to its activation and initiates a complex network of downstream signaling events that regulate diverse cellular processes.

**Figure 2 cells-14-01325-f002:**
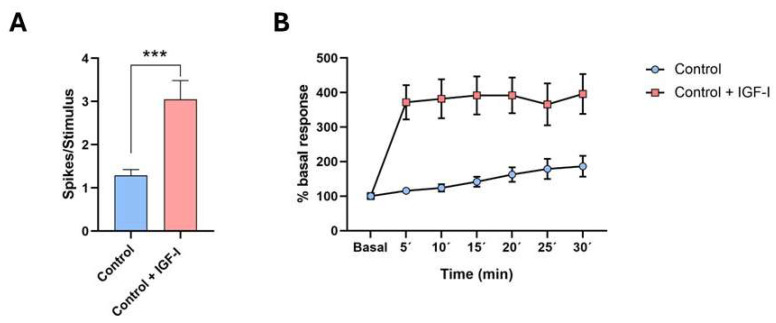
IGF-I enhances neuronal responses to whisker stimulation in the primary somatosensory cortex. (**A**) The plot shows the mean responses (spikes/stimulus) to whisker stimulation under control conditions and after 15 min of local IGF-I application (10 nM; 0.2 mL) in layer 2/3 neurons of the primary somatosensory cortex of mice. A significant increase in whisker responses was observed in the presence of IGF-I (***, *p* = 0.0002). (**B**) Temporal evolution of mean whisker responses following a train of whisker stimuli (8 Hz; 10 s at time 0), recorded from layer 2/3 neurons of the primary somatosensory cortex in mice. Under control conditions, the whisker response increased slowly; however, in the presence of IGF-I, a significant enhancement in response facilitation was observed (two-way ANOVA F(1.287 = 137.4) *p* < 0.0001). Modified from [53].

**Figure 3 cells-14-01325-f003:**
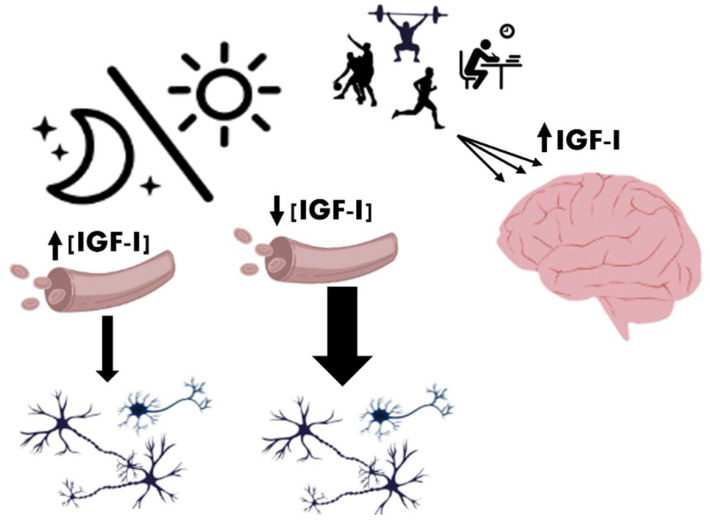
Circadian dynamics of IGF-I in relation to sleep and activity levels. During slow-wave sleep at night, GH secretion increases, leading to higher circulating IGF-I levels. However, due to reduced neuronal and physical activity, IGF-I brain uptake is limited. In contrast, during the day, despite lower systemic IGF-I levels, increased physical and neuronal activity enhances IGF-I entry into the brain via activity-dependent mechanisms. This inverse relationship suggests a dynamic balance between peripheral IGF-I levels and its central availability across the sleep–wake cycle.

## Data Availability

No new data were added or analyzed in this study. Data sharing is not applicable to this article.

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
