# Peer review of "Modulatory Action of Insulin-like Growth Factor I (IGF-I) on Cortical Activity: Entrainment of Metabolic and Brain Functions"

_cells, 2025, doi:10.3390/cells14171325_

Round 1
Reviewer 1 Report (Previous Reviewer 1)
Comments and Suggestions for Authors
The manuscript text is much improved and additional sections give more context and depth.
Reviewer 2 Report (Previous Reviewer 2)
Comments and Suggestions for Authors
The authors have revised the manuscript in accordance with the reviewer recommendations.
This manuscript is a resubmission of an earlier submission. The following is a list of the peer review reports and author responses from that submission.
Round 1
Reviewer 1 Report
Comments and Suggestions for Authors
The manuscript aims to summarize the role of insulin-like growth factor I (IGF-I) in modulating brain activity and its relevance to integrate metabolic processes and brain function. While the manuscript covers a relevant and interesting topic, the overall writing style detracts from its scientific impact and readability and requires significant improvement. Several issues with phrasing, grammar, and organization diminish the clarity and quality. See below for specific comments meant to help the authors make major changes to enhance the manuscript’s readability, accessibility, and professional presentation.
Major comments:
- Depth of mechanistic insight - The broader interaction between metabolism, IGF, and brain activity is not well articulated. While the review does address known actions of IGF-I on specific neuronal populations (such as cholinergic and orexinergic neurons), it would benefit from a more detailed mechanistic discussion. For example, the molecular pathways downstream of IGF-I receptor activation in cortical neurons, and how these relate to synaptic plasticity and electrophysiological changes, could be explored in greater depth.
- IGF-1 connection to metabolism - There is minimal coverage of the role of IGF-1 in metabolic regulation (such as glucose transport or mitochondrial function) and how that is related to brain activity. As this is one of the main topics that this review purports to summarize, the article appears to have a huge gap.
- Organizational structure - Many sections on IGF-1 and brain function appear to be poorly organized with either a superficial level of detail or incomplete coverage. The text wanders and contains vague collections of details pertaining to the section topics which do not provide a coherent or sufficient review of most topics. Introduce each main topic and then explore in increasing detail. Separate mechanisms and functional outcomes for clarity like synaptic plasticity vs. sleep regulation. At the end, address the translational and therapeutic implications, tying them back to the central theme. Clarify subheadings to help guide the reader and reduce redundancy.
3a) Lack of flow and logical transitions - Sections sometimes jump abruptly between ideas without adequate transition, making it challenging for the reader to follow the argument. For example: The manuscript moves from discussing hippocampal LTP to electrocorticogram changes after IGF-I manipulation without clear transitions or linking sentences.
3b) Poor paragraph structure - The following paragraph that starts with the following would benefit from improved flow, concise sentences, and linking the findings to their significance.
“The above findings indicate that IGF -I modulates intrinsic and synaptic properties …”
- Balance of evidence - The review primarily emphasizes positive, facilitatory roles of IGF-I. Including a critical assessment of conflicting or negative findings from the literature, or addressing limitations in the field (challenges in distinguishing direct versus indirect effects), would add balance.
- Scope of pathological conditions - The manuscript discusses the reduction in IGF-I levels in aging, diabetes, and Alzheimer’s disease in relation to cognitive and sleep deficits. Expanding this section with more specific clinical data or referencing interventional studies where IGF-I or it’s signaling pathway has been therapeutically targeted would strengthen clinical relevance.
- English, abbreviations and accessibility overview - Carefully proofread the manuscript or consider professional language editing to correct grammatical, typographical, and syntax errors. Simplify sentence structure and avoid jargon when possible, especially in the introduction and abstract. Improve logical flow by providing clear transitions and grouping related ideas coherently. Use consistent terminology and limit abbreviations to those essential for scientific communication. Minimize repetition and unnecessary qualifiers. Highlight key findings with concise, direct statements and avoid passive voice unless it improves clarity. See below for specific recommendations and examples:
6a) Some sentences are densely written and would benefit from editing for brevity and clarity. There are numerous grammatical mistakes, many typographical and formatting inconsistencies (e.g., inconsistent use of hyphenation such as "insulin -like grow factor I", which should read "insulin-like growth factor I").
6b) There are too many abbreviations (ACh, BF, CNS, ECoG, GH, etc.) for words used only a few times, try minimizing abbreviation use in the main text where possible. In several places, terms are introduced without sufficient explanation or are used inconsistently. Examples include: “ECoG” and “EEG” are both used when referring to electrophysiological measures; consistent terminology should be chosen, and brief explanations should be provided on first use.
6c) Improve the awkward wording in multiple places for improved comprehension. For example: Abstract: “Insulin -like grow factor I (IGF -I) is a neurotrophic factor controlling the growth and function of all the major types of brain cells.” And “IGF -I increases neuronal activity and facilitates synaptic plasticity in many brain regions.” While technically correct, this should be followed with a brief, concrete example or supporting statement for clarity.
6d) Redundancy and wordiness - Some points are repeated throughout the text, and verbosity is common. For instance: “This effect can be achieved through the direct effect of IGF-I on cortical neurons, or through activation of cholinergic and orexinergic neurons, facilitating information processing in the CNS.” The phrase “facilitating information processing in the CNS” is repeatedly used without further explanation or specificity. Unnecessary qualifiers such as “the above findings indicate that” or “in fact” do not add scientific value and should be omitted or replaced with more precise language.
6e) Overuse of passive voice - Passive voice is frequent, sometimes obscuring the point and making the text less engaging. For example: “A reduction of IGF -I circulating level and the input to the CNS, as occur in healthy aging or in pathological conditions such as diabetes or Alzheimer’s disease, may be responsible for cognitive deficits as well as poor sleep quality that occur in these patients.”
Minor comments:
- Title - The title should include a colon vs a period
- Figures - Think about what each figure is trying to communicate. Figure 2 does not tell us any additional information. Make sure the figure axes are labeled and descriptions are self-contained for clarity.
- Conclusion - The conclusion section summarizes the main findings and their significance but could be strengthened by a brief mention of future directions or outstanding questions in the field.
- Check reference formatting for uniformity.
see above for major revision suggestions
Reviewer 2 Report
Comments and Suggestions for Authors
In this review, the authors underlying the rule of Insulin-like grow factor I (IGF-I), summarized recent findings about the interaction between metabolism, IGF-I and brain activity. The data suggests that IGF-I may be one of the key factors to produce integration between the metabolic and neuronal systems optimizes energy utilization throughout the light/dark cycle.
In general, the work is well articulated, despite this I still have some questions.
Questions:
In some places the text is redundant.It is also recommended to go into more detail on some points, in particular, to explain why IGF-I sometimes inhibits synaptic transmission; and what concrete clinical and/or therapeutic implications could derive. Describe the limitations of the studies of animal models; in this regard, Figure 1A shows that IGF-I local application on the S1 cortex enhances tactile responses in mice. IGF-I also increases the size of receptive fields in the S1 cortex of rats but has no effect on the somatotopic map. I didn't understand why the authors included this figure. Please explain.
The authors describe that IGF-I signaling shows circadian rhythm, please explain if there is scientific evidence between IGL-1 and other stress-hormones.
There is a lack of general discussion or an expansion of the conclusions which are poor. Extend the conclusions and provide a critical comparison between the data presented and potential implications and future directions.
Finally, I suggest creating a figure that summarizes the intracellular pathways involved in metabolic and brain activity.
Comments on the Quality of English LanguageMany grammatical and syntactical errors in English present in the text.